# Imaging of Thromboinflammation by Multispectral ^19^F MRI

**DOI:** 10.3390/ijms26062462

**Published:** 2025-03-10

**Authors:** Sebastian Temme, Patricia Kleimann, Zeynep-Büsra Tiren, Pascal Bouvain, Arthur Zielinski, William Dollmeyer, Sarah Poth, Juliana Görges, Ulrich Flögel

**Affiliations:** 1Department of Anesthesiology, Faculty of Medicine, University Hospital, Heinrich-Heine-University, 40225 Düsseldorf, Germany; zetir100@uni-duesseldorf.de (Z.-B.T.); arthur.zielinski@uni-duesseldorf.de (A.Z.); widol100@uni-duesseldorf.de (W.D.); juliana.goerges@hhu.de (J.G.); 2Experimental Cardiovascular Imaging, Institute of Molecular Cardiology, Faculty of Medicine, University Hospital, Heinrich-Heine-University, 40225 Düsseldorf, Germany; patricia.kleimann@uni-duesseldorf.de (P.K.); pascal.bouvain@uni-duesseldorf.de (P.B.); sarah.poth@hhu.de (S.P.); floegel@uni-duesseldorf.de (U.F.)

**Keywords:** thromboinflammation, ^19^F MRI, multispectral ^19^F MRI, multicolor ^19^F MRI, fibrin, FXIIIa, gpIIb/IIIa, neutrophils

## Abstract

The close interplay between thrombotic and immunologic processes plays an important physiological role in the immune defence after tissue injury and has the aim to reduce damage and to prevent the spread of invading pathogens. However, the uncontrolled or exaggerated activation of these processes can lead to pathological thromboinflammation. Thromboinflammation has been shown to worsen the outcome of cardiovascular, autoinflammatory, or even infectious diseases. Imaging of thromboinflammation is difficult because many clinically relevant imaging techniques can only visualize either inflammatory or thrombotic processes. One interesting option for the noninvasive imaging of thromboinflammation is multispectral ^19^F magnetic resonance imaging (MRI). Due to the large chemical shift range of the ^19^F atoms, it is possible to simultaneously visualize immune cells as well as thrombus components with specific ^19^F tracer that have individual spectral ^19^F signatures. Of note, the ^19^F signal can be easily quantified and a merging of the ^19^F datasets with the anatomical ^1^H MRI images enables precise anatomical localization. In this review, we briefly summarize the background of ^19^F MRI for inflammation imaging, active targeting approaches to visualize thrombi and specific immune cells, introduce studies about multispectral ^19^F MRI, and summarize one study that imaged thromboinflammation by multispectral ^19^F MRI.

## 1. A Brief Introduction to Immune Thrombosis and Thromboinflammation

Thrombotic processes, as part of innate immune responses, play an important physiological role in immune defence, e.g., stopping bleeding in the case of external injuries and preventing or reducing the spread of pathogens, thereby protecting the wound area from infection without causing major collateral damage to the organism. This physiological interaction between immune cells, platelets, and blood-clotting molecules is known as immune thrombosis [1]. However, if an uncontrolled process or a greatly increased reaction occurs, the immune thrombosis changes into a pathological form known as thromboinflammation, which can significantly influence the pathology of multiple disease states, especially in cardiovascular diseases (e.g., abdominal aortic aneurysms) [2]. Excessive thromboinflammation can cause serious damage to the vessel wall which plays a crucial role in atherosclerosis. Chronic inflammation of the arterial walls leads to the formation of atherosclerotic plaques and the rupture of plaques results in thromboinflammatory reactions activating platelets and causing the formation of blood clots, leading to myocardial infarction or stroke, while the resulting vascular damage exacerbates the progression of the disease [3].

A crucial aspect of these thromboinflammatory processes is associated with endothelial dysfunction, which can be triggered by oxidative stress (reactive oxygen species [ROS]) [4], pathogenic microorganisms [5], or proinflammatory molecules (e.g., lipids or cytokines) [6]. Damage to the endothelium leads to the increased expression of adhesion molecules for platelets, monocytes, or neutrophils, such as von Willebrand factor (vWF), P-selectin, or intercellular cell adhesion molecule 1 (ICAM-1) [7]. Furthermore, there is an increase in the secretion of vasoactive substances and proinflammatory cytokines, e.g., tumour necrosis factor α (TNFα) [8] or interleukin-6 (IL-6) [9], which contribute to the recruitment of platelets and leukocytes and lead to a procoagulatory state [10]. In conjunction with fibrinogen as an acute-phase protein, increased IL-6 secretion causes an elevated thrombogenic potential by activating platelets and releasing tissue factor (TF). The expression of vWF and P-selectin on damaged and/or activated endothelial cells supports the binding and rolling of platelets and leukocytes [11,12].

Stable platelet adhesion is, for example, mediated by fibrinogen-integrin αIIbβ3 complexes that bind to ICAM-1 on the endothelium. Adherent platelets secrete numerous bioactive substances that alter the chemotactic and adhesive properties of endothelial cells [13]. For instance, platelets secrete IL-1 which induces TF expression on the endothelium and leads to the activation of coagulation factors FVII and FX and ultimately to the formation of thrombin. Thrombin cleaves fibrinogen to form fibrin and several protease-activated receptors on the surface of endothelial cells, platelets, and leukocytes, thereby continuing the thromboinflammatory process. In addition to the formation of soluble fibrin monomers, thrombin causes the activation of coagulation factors such as factor XIII. Activated factor XIII (FXIIIa) is a transglutaminase that produces cross-links between proteins. The FXIIIa-mediated formation of covalent bonds between fibrin monomers in the early phase of thrombus formation leads to stabilization of the fibrin network and finally to contraction of the thrombi [14]. The expression of P-selectin on endothelial cells or adherent activated platelets in turn induces the binding of leukocytes, mainly through the interaction of the macrophage-1 antigen (MAC-1) with glycoprotein (GP) Ib-GPV-GPIX and fibrinogen–integrin αIIbβ3 complexes. In addition to the interaction of leukocytes with platelets and endothelial cells, monocytes also secrete TF and proinflammatory cytokines or chemokines (e.g., IL-1β, IL-6, TNFα, CCL2, and CXCL2) to ensure the recruitment of additional platelets and leukocytes. Neutrophils are known to release of so-called neutrophil extracellular traps (NETs)—net-like structures made of fine chromatin fibres and intracellular granule contents—which also represent an important interface or interaction mechanism of inflammation and coagulation [15,16]. Furthermore, neutrophils as well as platelets secrete proteolytic enzymes like matrix-metalloproteinases (MMP) -2 or -9 to degrade the extracellular matrix and improve thrombus formation. A summary of the key characteristic features of thromboinflammation is shown in Figure 1.

One of the primary causes of death to date are cardiovascular diseases (CVDs). The clinical importance of inflammation associated with CVDs was shown in the 2017 *Canakinumab Antiinflammatory Thrombosis Outcome Study* (CANTOS) [17]. Patients with a previous myocardial infarction received an antibody-inhibiting IL-1β (Canakinumab; 50 mg, 150 mg or 300 mg) or a placebo. The intervention group showed a significantly reduced incidence of repetitive atherothrombotic events (non-fatal myocardial infarction (MI), non-fatal stroke, or cardiovascular death) and a reduction in the C-reactive protein level. In particular, the group that received 150 mg canakinumab showed a reduction in primary cardiovascular end-points (nonfatal MI or stroke, or cardiovascular death; hazard ratio versus placebo, 0.85, *p* = 0.02075). Therefore, anti-IL-1β treatment may pose a potential new target for the treatment of CVDs and shows the clinical relevance of understanding thromboinflammatory processes. However, this study also showed the drawback of an anti-inflammatory IL-1β therapy. Canakinumab treatment was associated with a higher incidence of fatal infections compared to the control group, which all together balanced the overall mortality in both groups [17].

In summary, thromboinflammatory processes refer to the interaction between a variety of humoral and cellular signalling pathways of the innate immune defence and the coagulation cascade. Due to the complexity of these processes, only the pathological consequences of excessive thromboinflammatory processes, such as tissue hypoperfusion, edema formation, and necrosis, can be detected with available clinically relevant non-invasive imaging techniques, which prevents early prophylactic treatment. In order to better understand these processes, it would be desirable if the close interaction between thrombus formation and immune cell activation or recruitment could be visualized using non-invasive imaging methods. To date, the most relevant imaging methods are ultrasound, CT (computer tomography), SPECT (single photon emission computed tomography), PET (positron emission tomography), and MRI (magnetic resonance imaging). Whereas ultrasound, CT, and anatomical proton MRI are often used to obtain information about structural changes in the tissue, to monitor immune responses, and the imaging of thrombus components has been conducted with PET, SPECT, or contrast enhanced MRI.

Numerous probes for PET/SPECT imaging have been designed and utilized for imaging of inflammation such as ^18^F-FDG (FDG = Fluorodeoxyglucose; taken up by macrophages, leukocytes) [18], ^18^F-mannose [19], ^68^Ga-DOTA-TATE to target the somatostatin receptor subtype 2 [20], ^68^Ga-Fucoidan (which binds to P-selectin) [21], and ^68^Ga-Pentixafor which recognizes CXCR4 [22]. Furthermore, thrombi have been detected via fibrin-binding peptides (FBP) (e.g., ^68^Ga-FBP14/^111^In-FBP15 [23], probes that target factor XIII [24], or the ligand-induced binding site of gpIIb/IIIa on activated platelets [25]. The major advantage of PET and SPECT imaging is the very high sensitivity. However, the spatial resolution is limited (3–8 mm^3^) and the patient is exposed to ionizing radiation [26]. The focus of this review is multispectral ^19^F MRI and, therefore, we cannot cover the broad range of approaches for PET/SPECT imaging, but we refer the reader to some excellent reviews about this topic [26,27,28,29]. However, these imaging methods are not able to visualize several target structures together, as PET or SPECT, for example, only detect one tracer at a time. Therefore, information about inflammation, certain immune cells, and thrombus formation cannot be collected simultaneously.

One way of visualizing several components of thrombotic and inflammatory processes is multispectral ^19^F MRI, which uses fluorine-containing substances, e.g., perfluorocarbon nanoemulsions (PFCs), to enable the background-free detection of a variety of cells and structures involved in the development and progression of thromboinflammatory processes. In the next section, the basic principles and some possible applications of multispectral ^19^F MRI are explained in more detail.

## 2. Imaging of Inflammation by Combined ^1^H/^19^F-MRI

Imaging of inflammatory processes by conventional ^1^H-based MRI is challenging because, in the early phase in particular, inflammation does not result in physical alterations that can be exploited to generate MR contrast. To this end, several indirect MRI methods have been utilized to gain information on inflammation such as the generation of tissue edema which can be detected by T2 mapping, or the application of gadolinium labelled small molecules that accumulate in inflamed tissue due to an impaired endothelial barrier. One of the hallmarks of inflammation is the migration of immune cells from the blood into the inflamed tissue. In many cases, the first immune cells that accumulate in diseased tissue are neutrophils, which are followed by classical and nonclassical monocytes. At later time points, lymphocytes such as T cells or B cells also accumulate in the inflamed lesion.

The visualization of immune-cell infiltration and accumulation is not feasible with classical ^1^H MRI. To overcome this, contrast agents such as iron oxide nanoparticles have been developed that are avidly taken up by circulating monocytes and local macrophages and which have been utilized in multiple preclinical but also clinical studies to visualize inflammation [30,31,32,33]. Small superparamagnetic iron oxide nanoparticles (SPIOs) belong to the group of negative MR contrast agents, because iron oxide disturbs the local magnetic field which results in a rapid decay in the signal intensity of T2-weighted anatomical ^1^H images [34,35]. The main advantage of SPIOs is the very high sensitivity, which enabled the single-cell tracking of patrolling monocytes in brain of mice [36]. On the other hand, the SPIO-mediated destruction of the ^1^H signal can impair the precise anatomical localization and the quantification is difficult because there is no linear correlation between iron concentration and signal loss. Furthermore, detection of SPIO-labelled monocytes and macrophages in the lungs (which normally have a very low proton density), or at the interphase between different tissues such as the vascular wall can be challenging.

An alternative technology for imaging immune-cell infiltration by MRI is based on fluorine 19 (^19^F). ^19^F has a gyromagnetic ratio similar to ^1^H, a natural abundance of 100%, and is nearly absent from biological tissue. There are low amounts of ^19^F in bones and teeth, but there, it is found in the form of fluorides which have a very short T1 relaxation and are therefore invisible to conventional imaging methods.

### ^19^F MR Imaging of Monocytes and Macrophages

The potential of ^19^F MRI for biomedical application has already been suggested in the first manuscript that showed the principle feasibility of ^19^F MRI in 1977 [37]. Holland et al. visualized a solution of NaF and perfluorotributylamine (FC43) by ^19^F MRI and suggested that this technology might be interesting for the observation of fluorinated drugs and the assessment for perfluorocarbons as artificial blood substitutes [37]. Studies in the 1980s showed that perfluorocarbon nanoemulsions could be applied for the visualization of inflammatory lesions because it was found that intravenously injected PFOB-PFCs or Fluosol-DA (A mixture of perfluodecalin and perfluortributylamin) did not only accumulate in the liver and spleen, but also in intraperitoneal bacterial abscesses and tumours [38,39,40]. Interestingly, in these early studies, imaging of the PFCs was not conducted by MRI, but by ultrasound [38,40] and computed tomography (CT) [39]. In 1985, Longmaid et al. utilized ^19^F MRI to investigate the localization of PFTA-PFCs (PFTA = perfluorotributylamine) that were i.v. injected into rats with Walker-256 sarcomas or intraperitoneal bacterial abscesses [41]. Here, PFTA-PFCs were found in the liver, but also in thigh tumours and abscesses. Histological analyses (conventional light microscopy and electron microscopy) revealed that the PFCs in the rim of the tumours were predominantly found in macrophages.

In 2005, Ahrens et al. published a landmark paper where dendritic cells were labelled ex vivo with PFCs, injected into the leg, and their trafficking towards lymph nodes was followed by ^1^H/^19^F MRI [42]. Three years later, our group then showed the feasibility of intravenously injected PFCs to image the accumulation of monocytes and macrophages within the heart and brain after ischemia–reperfusion injury [43]. Here, we systematically investigated the cellular uptake and localization of i.v.-injected PFCs. Flow cytometry and ^19^F MRI of blood immune cells separated by ficoll gradient density centrifugation revealed that PFCs are predominantly taken up by circulating monocytes. Histological analyses also verified that the PFC signal in the heart strongly colocalized with cardiac macrophages. However, some PFC uptake was also found by neutrophils and B-cells, whereas T cells did not internalize the PFCs.

Many follow up studies showed that the majority of the PFCs are found in monocytes and macrophages and to a lesser extend in other phagocytic immune cells. For example, Ebner et al., performed ^19^F MRI of the lungs after pulmonary instillation of lipopolysaccharide (LPS) [44]. Although neutrophils outnumbered monocytes/macrophages in the lungs after LPS treatment, the strongest ^19^F signal was found in macrophages. Of note, apart from ^19^F MRI, early pulmonary inflammation has also been detected by ^1^H MRI (edema) and ^3^He MRI which can be used to visualize ventilation [45,46]. Similar results regarding ^19^F MRI were obtained for atherosclerosis [47], inflammatory bowel disease [48], myocardial infarction in pigs [49], or viral myocarditis in mice [50]. However, some studies do also show that under certain circumstances other cell types such as neutrophils [51] or even a distinct stem cell population of the heart [52] can be predominantly labelled after the intravenous application of PFCs.

## 3. Active Targeting of Perfluorocarbon Nanoemulsions

In the previous section, we described that conventional perfluorocarbon nanoemulsions are predominantly taken up by circulating and local phagocytes such as monocytes, macrophages but also neutrophils. Although it has been shown that this approach is suitable for imaging of inflammation in a variety of different disease models, it is limited to the phagocytic properties of certain cell types. Therefore, an active targeting of PFCs is necessary to direct PFCs to cells/structures that cannot be labelled by conventional PFCs. To achieve an active targeting, the surface of the PFCs is functionalized with ligands, such as antibodies, single-chain antibodies, nanobodies, peptides, or other small molecules like sugars or cytokines that facilitate the binding of the nanoparticles to the desired structure or cell type. These ligands are often covalently attached to the nanoprobes via maleimide-thiol-, carboxyl-carbodiimide-, or NHS (N-hydroxysuccinimide)-ester-lysine-reactions [53]. Another way of binding is the biotin-streptavidin system, a non-covalent method which involves biotinylated targeting molecules that strongly bind to streptavidin- or avidin-coated nanoparticles [54]. Furthermore, multivalent probes, which combine multiple targeting peptides, could increase binding specificity and affinity [55].

What is also important is that the phagocytic uptake of PFCs by monocytes/macrophages and other cell types has to be reduced. The most widely used strategy to impair the phagocytic uptake is the modification of the nanoparticle surface with poly-ethylene glycol (PEG). PEG reduces the attachment of serum proteins (e.g., apolipoproteins, complement, and immunoglobulins) to the particle surface and therefore the formation of the nanoparticle corona [56]. These serum proteins (e.g., complement proteins), are recognized by phagocytic cells and facilitate the internalization of the nanoparticles [57]. However, one drawback of PEG is that it is immunogenic and induces the generation of anti-PEG antibodies which can either neutralize PEGylated nanoparticles or lead to anaphylactic reactions [58]. Therefore, alternative methodologies have been developed to confer stealth properties to nanoparticles, such as the use of polyphosphoester [59], CD47, or peptides derived from CD47 [60,61].

Interestingly, many earlier studies that have utilized targeted PFCs did not perform ^19^F MRI, but used them for imaging by X ray [62], ultrasound [63], near-infrared microscopy [64], or T1-weighted MRI, because the PFCs were decorated with a high payload of gadolinium [65,66,67]. Active targeting of PFCs has enabled the specific visualization of FXIIIa activity in early thrombi [68,69], activated platelets [70], fibrin [71], cells that express synthetic cargo-internalization receptors [72], epicardial derived stem cells, and activated cardiac fibroblasts [73,74] or neutrophils [75].

Taken together, the application of unmodified PFCs enables the imaging of (predominantly) monocytes and macrophages which accumulate in inflamed lesions. Modification of the surface of PFCs with PEG to impair the endocytic uptake by phagocytic cells in combination with the attachment of specific targeting ligands makes it possible to visualize structures, cells, and processes which could not be specifically visualized by the conventional approach. The following two sections describe in more detail the ^19^F MRI-based imaging of thrombus formation and the trafficking of neutrophils from the bone marrow to the infarcted heart.

### 3.1. Imaging of Thrombi by Targeting of FXIIIa, Activated Platelets, and Fibrin

As mentioned above (Section 1), the activated form of FXIII cross-links the fibrin network during the early phase of the thrombus formation process by formation of lysyl-bonds [76,77]. FXIIIa does also cross-link several proteins to fibrin, such as α2-antiplasmin. Due to the fact that FXIIIa is limited to the early phase of the thrombus formation process, Miserus et al. utilized a 14 amino acid peptide from α2-antiplasmin to develop an MRI-tracer to detect early thrombus formation [78]. To this end, they modified the peptide with gadolinium-chelates that resulted in signal enhancement in thrombi by imaging with T1-weighted proton MRI sequences.

We adopted this approach and modified the α2-antiplasmin peptide to facilitate the conjugation to the surface of PFCs. The C-terminus of the peptide was extended by a short GKG-peptide spacer where the ε-amino group of the lysine side chain was functionalized with carboxyfluoresceine and a terminal cysteine. This α2^AP^-peptide was conjugated to cholesterol-PEG_2000_-maleimide to generate cholesterol-PEG_2000_-α2^AP^ (Chol-PEG-α2^AP^). The SH-group of the C-terminal cysteine reacts with the maleimide group to form a stable thioether. Chol-PEG-α2^AP^ was then incubated with preformed PFCs at room temperature which leads to the spontaneous insertion of the cholesterol moiety of Chol-PEG-α2^AP^ into the lipid layer of the PFCs, in a process called sterol-based post-insertion (SPIT) which finally led to the generation of α2^AP^-PFCs. SPIT was originally developed to functionalize liposomes [79] and enables the insertion of labile ligands into the lipid layer of liposomes or PFCs. During manufacturing, the pre-emulsions are processed at a high pressure of 1000 bar which is also associated with the generation of heat that can destroy labile ligands.

As control, a peptide was conjugated to PFCs where the glutamine on position 3 was converted to alanine (Q3A). This peptide shows an about 10-fold lower accumulation in the thrombus, because Q3 is mainly utilized by FXIIIa for cross-linking to the fibrin network. The specificity of this targeting approach was first evaluated by in vitro-generated human thrombi and subsequent ^19^F MRI measurements. Here, it turned out that PFCs labelled with α2^AP^ showed a much higher accumulation in acute in vitro-generated thrombi than those functionalized with Q3A. To evaluate the functionality under in vivo conditions, we utilized a mouse model of deep venous thrombosis where thrombus formation was induced by application of a small piece of FeCl_3_-soaked filter paper on the top of the external side of the vessel wall of the inferior vein. Intravenous injection of α2^AP^-PFCs prior to the thrombus induction resulted in a very strong accumulation of the α2^AP^-PFCs and therefore enabled an unequivocal identification of the thrombus (Figure 2).

Imaging of thrombi by α2^AP^-PFCs enables the identification of thrombi during the very early phase but does not allow the visualization of more advanced thrombi. The reason is that the activity of FXIIIa is restricted to the early phase of the thrombus formation and the enzymatic activity rapidly declines after the thrombus has stabilized and contracted. However, thrombus formation can also be visualized via activated platelets or targeting the formed fibrin network. Activation of platelets is associated with multiple alterations and one important aspect is the change in the conformation of the integrin gpIIb/IIIa from a low- into a high-affinity state. Several years ago, KH Peter and his group developed a single-chain antibody which is directed against the activated form of gpIIb/IIIa (^scFV^LIBS) [80,81]. The suitability of this single-chain antibody for visualization of activated platelets has been validated by multiple different imaging modalities, like ultrasound, PET, fluorescence-based methods, and MRI after labelling of iron oxide nanoparticles [82,83]. More recently, we also utilized ^scFV^LIBS to image activated platelets in murine FeCl_3_-induced deep venous thrombi by combined ^1^H/^19^F MRI [70].

Several years ago, phage display screening identified peptides with a high specificity for fibrin. Derivates of these peptides (e.g., EP-2104R) were utilized for in vivo visualization of thrombi in multiple preclinical studies [84,85,86], but also in clinical studies [87]. More recently, we utilized a modified version of the peptide moiety of EP-2104R to functionalize PFCs and to image thrombi in FeCl_3_-induced deep venous thrombi and also in thrombi formed in the lungs and the cardiac vasculature in HypoE mice that were treated with a high fat/cholesterol diet (see Section 4.2).

### 3.2. Visualization of Neutrophil Trafficking

Neutrophils are a crucial component of the innate immune system. They are produced in the bone marrow by hematopoietic stem cells. Humans, generating approximately 10^11^ neutrophils daily [88], differ from mice, where the daily production is around 10^7^ cells [89]. Under normal, non-inflammatory conditions, most neutrophils are retained within the bone marrow, continuously being released into the bloodstream in an inactive state, where they patrol the blood vessels. However, during infection or tissue injury, a process called emergency granulopoiesis is activated and high amounts of neutrophils are released from the bone marrow into the circulation [90]. Furthermore, neutrophils become activated and are recruited to the site of inflammation. After infiltration into the inflamed tissue, neutrophils rapidly engulf infectious microorganisms to prevent spreading of the infection. Neutrophils do also remove cell debris—not only during infection—but also in situations of sterile inflammation caused by trauma or ischemic tissue injury. Given their role as first responders of the immune system, there is growing interest in understanding their natural behaviour and impact during inflammatory processes.

As mentioned in Section 2, the passive uptake of PFCs by phagocytic cells in ^19^F-based approaches lacks specificity for neutrophils. In complex inflammatory environments where multiple cell types are involved, neutrophils are often only weakly labelled compared to the highly phagocytic monocytes and macrophages [43,44]. This was seen, for example, by Ebner et al. during an acute lung injury model where PFC-labelled monocytes and neutrophils entered the lung [44]. To enable selective visualization of neutrophils using ^19^F MRI, PFCs must be equipped with ligands that target neutrophil-specific cell-surface receptors while simultaneously minimizing phagocytic uptake by monocytes and macrophages. In 1999, Mazzucchelli et al. identified peptides that specifically bind to human neutrophils [91]. Later, our group demonstrated that one of these peptides (hNP) binds to human CD177, a GPI-anchored surface protein specifically expressed by neutrophils. In most individuals, approximately 50–60% of the cells are positive for CD177 [75], although there is a large range from 0 to 100% [92]. However, it was also observed that hNP does not bind to murine CD177. Nearly two decades later, Miettinen et al. identified a peptide that targets murine CD177 [93]. Interestingly, unlike in humans where CD177 is found on 50–60% of the neutrophils in the blood, 100% of blood neutrophils in different mouse strains (e.g., C57BL/6J, Balb/c, and NMRI) express CD177 on the cell surface [75]. Using these peptides, we recently developed PFCs that are specifically targeted to either human or murine neutrophils (^hNP^PFCs and ^mNP^PFCs, respectively). Both targeting peptides were modified with a triple glycine linker and a terminal cysteine, allowing conjugation to maleimide groups on the PFC surface. Additionally, PFCs were functionalized with polyethylene glycol (PEG) to reduce the non-specific uptake by monocytes and macrophages [68,94,95]. When ^NP^PFCs were incubated with blood-derived immune cells, they showed highly specific labelling of neutrophils, with minimal uptake by monocytes or lymphocytes. Of note, there is some variability in the uptake of ^NP^PFCs which depends on the cell type or whether the cells are derived from the blood or from an inflammatory lesion. In general, there is a 5- to 50-fold higher cellular uptake of ^NP^PFCs by neutrophils compared to monocytes, macrophages, or lymphocytes [75]. Enhanced neutrophil uptake of ^NP^PFCs compared to unspecific control PFCs (^Con^PFCs) was also confirmed using ^19^F MRI. During inflammation, neutrophils are rapidly released from the bone marrow into the bloodstream and migrate to the site of injury. To visualize neutrophil trafficking from the bone marrow to inflammatory lesions using combined ^1^H/^19^F MRI, we pre-labelled neutrophils in the bone marrow by repetitive injection of ^mNP^PFCs prior to myocardial infarction [75]. Whole-body 3D ^1^H and ^19^F MRI conducted before and one day after myocardial infarction revealed neutrophil egress from the bone marrow and subsequent infiltration into the infarcted heart (Figure 3). Following myocardial infarction, the ^19^F signal in the femur (which represents bone marrow neutrophils) significantly decreased, while a corresponding ^19^F signal appeared in the infarcted heart (Figure 3 upper panel and lower left panel).

## 4. Multispectral ^19^F MRI of Thromboinflammation

Multispectral ^19^F MRI is an advanced imaging technique that allows the simultaneous visualization and tracking of multiple molecular targets within the same region of interest. This method utilizes the unique properties of fluorine-19 (^19^F) nuclei in magnetic resonance imaging. The negligible presence of ^19^F in the human body provides accurate and sensitive background-free detection and visualization of fluorine compounds. One further advantage of this imaging technique is the wide chemical shift range of ^19^F, which is in the range of 300 to 400 ppm and enables the use of multiple probes with distinct ^19^F signatures. Considering that signals differing by at least 5 ppm are easy to distinguish, up to 70 theoretical “channels” are available for multispectral ^19^F imaging. This enables the identification of multiple fluorinated tracers with well-separated spectral peaks such as perfluorocarbon nanoemulsions with individual spectral signatures.

For specific cell tracking or labelling of structures by multispectral ^19^F MRI, the surface of the PFC-nanoprobes must be modified to achieve selective binding to the desired target structure, or to label cells ex vivo with ^19^F tracer and then reintroduce the cells into an organism to track them in vivo [96].

### 4.1. Studies That Utilize Several Fluorinated Tracers for Multispectral ^19^F MRI

Multispectral ^19^F MRI has shown promise in various applications such as molecular or cancer imaging and cell tracking (Table 1). The first study on multicolor ^19^F MRI was published in 2007 and showed PFC-loaded nanoprobes (PFOB- [perfluorooctyl-bromide] and PFCE-PFCs) for in vivo tracking of stem/progenitor cells [96]. A few years later, our group implemented a technology for the simultaneous imaging of various mixtures of different PFC emulsions ex vivo and in vivo [97]. We used a multi chemical shift selective (MCSS) RARE sequence for artefact-free ^19^F MR imaging of perfluorocarbons (PFC) with complex spectra. In a further study, a method was developed that enables the simultaneous visualization of different PFCs with a significant reduction in acquisition time and reduced chemical shift artefacts that are normally associated with complex ^19^F MR spectra [98]. The group of Akazawa et al. already used three different types of PFC-encapsulated silica nanocarriers with distinct chemical shifts in the ^19^F peaks and showed the principle feasibility for multicolor in vivo imaging by subcutaneous injection of these ^19^F tracers into the back of the mice [99].

Further studies have demonstrated successful multicolor ^19^F MRI in living mice, visualizing different nanoprobes simultaneously. PFCE and PERFECTA (derived from “suPERFluorinatEdContrasT Agent”) were used to track the movement and activity of mononuclear cells in mice by ^19^F MRI [100]. Croci et al. utilized two different PFCs for multispectral ^19^F imaging of spatially and temporally distinct tumour-associated microglia and macrophages (TAMs) in radiotherapy-recurrent murine gliomas [101]. It was also possible to develop stable and clinically applicable labels for tracking two populations of primary human dendritic cells simultaneously [102].

Multicolor ^19^F MRI has shown promise in characterizing various cardiovascular diseases and tracking immune cell populations in myocardial infarction. Three PFC-probes were used to monitor monocytes, fibrin, and FXIIIa, providing detailed insights into myocardial infarction progression over two weeks by multicolor ^19^F MRI [103] (see Section 4.2). Table 1 summarizes studies the up to now utilized different ^19^F tracers such as perfluorocarbons, but also other ^19^F labelled molecules for multispectral ^19^F MRI (see below).

**Table 1 ijms-26-02462-t001:** Studies about multispectral ^19^F MRI.

	Brief Summary of the Multispectral ^19^F Approach	^19^F Tracer	Reference
1	In vivo tracking of human hematopoietic stem/progenitor cells. The labelled cells were injected in the right and left thigh of adult C57/BL6 mice.	PFCs: PFOB and PFCE	Partlow et al., 2007 [96]
2	Multi chemical shift selective (MCSS) RARE sequence for artefact-free ^19^F MR imaging of two different PFCs simultaneously in vitro and in vivo. Assessment of the washout kinetics from liver and spleen after intravenous injection of a mixture of PFOB- and PFCE-PFCs into male C57BL/6 mice.	PFCs: PFOB and PFCE	Jacoby et al., 2014 [97]
3	Labelling and simultaneous visualization of two populations of blood derived human plasmacytoid and myeloid dendritic cells. The labelled cells were injected intramuscularly into the leg of a mouse.	PLGA-nanoparticles: PFCE and PFO	Srinivas et al., 2015 [102]
4	Preparation of silica nanoparticles that encapsulate several perfluorocarbons. The in vivo suitability of this approach was shown by subcutaneous injection of the nanoparticles into the back of mice followed by multicolor ^19^F imaging.	Silica-nanoparticles: PFCE, TPFBME and PFTBA	Akazawa et al., 2018 [99]
5	PFCE and PERFECTA were emulsified with a nonionic surfactant (Pluronic F68). These nanodroplets were used to label and track the movement and activity of mononuclear cells in C57BL/6 mice after inhibition of the colony-stimulation factor-1 receptor by multicolor ^19^F MRI. Nanoemulsions were taken up by mononuclear cells, including neutrophils and monocytes.	Pluronic nanoemulsions: PFCE and PERFECTA	Chirizzi et al., 2019 [100]
6	Poly(lactic-co-glycolic acid) (PLGA) nanoparticles loaded with two fluorocarbons (PFCE and PERFECTA) for multicolor ^19^F MRI. These nanoparticles are composed of fractal blocks that contain PFCE and PERFECTA. Degradation of the PLGA results in changes in T1- and T2-relaxation of the perfluorocarbons which could be used for imaging of nanoparticle degradation in vivo.	PLGA-nanoparticles: PFCE and PERFECTA	Koshkina et al., 2019 [104]
7	Development of a method that enables the simultaneous visualization of different PFCs with a significant reduction in acquisition time and chemical shift artefacts in connection with complex ^19^F MR spectra. The ^19^F signal was also measured in mice (male C57BL/6) after intramuscular and intravascular injection of fluorine probes.	PFCs: PFCE and PFOB	Schoormans et al., 2020 [98]
8	Utilization of inhalable fluorinated anesthetics (halothane, isoflurane, sevoflurane, and fluroxene) in combination with water-soluble molecular cages (hosts) for ^19^F-GEST (guest exchange saturation transfer) for the simultaneous detection of micromolar concentrations of two targets.	Fluorinated anesthetics as guest: Halothane, Isoflurane, Sevoflurane, and Fluroxene	Shusterman-Krush et al., 2021 [105]
9	PFC-based multicolor ^19^F MRI probes for visualizing FXIIIa, fibrin, and monocytes/macrophages. PFCs with PFCH and PFTBH were PEGylated and functionalized with peptides against FXIIIa (^α2AP^PFCH) and fibrin (^fbn^PFTBH). Conventional PFCE-PFCs were utilizted to image phagocytic monocytes and macrophages. HypoE mice were subjected to a high-fat diet and a combination of ^α2AP^PFCH, ^fbn^PFTBH, and PFCE-PFCs were applied to visualize thromboinflammation in the lungs and the heart.	PFCs: PFCE, PFOB and PFCH	Flögel et al., 2021 [103]
10	Activatable ^19^F MRI probes for simultaneous in vivo detection and deep-tissue imaging of reactive oxygen species (ROS; O_2_^•–^) and reactive nitrogen species (RNS; ONOO^–^) in mice with drug-induced acute kidney injury. ROSP-1 and RNSP-2 contain paramagnetic gadolinium which reduces T1- and T2- relaxation of the fluorinated groups. ROS and RNP lead to the release of fluorinated groups from the ROSP-1 or RNSP-2 molecules and therefore the separation from the gadolinium which leads to a strong ^19^F signal increase.	Activatable ^19^F MRI molecular probes: ROSP-1 and RNSP-2	Li et al., 2021 [106]
11	Inorganic small fluoride nanoparticles (~10 nm; CaF_2_ and SrF_2_) were doped with the lanthanide samarium (sm) which significantly decreased the T1 relaxation and therefore increased the ^19^F signal. Sm:CaF_2_ nanoparticles were further coated with lipids and phospholipids to enhance the water solubility. Sm:CaF2 that were further modified with lactosyl-PE (LPL-Sm:CaF_2_) revealed an increased uptake by macrophages and showed an enhanced accumulation in lymph nodes and within an inflammatory hot spot of mice. Multiplex ^19^F MRI displayed an increased accumulation of LPL-Sm:CaF_2_ in lymph nodes of inflamed mice after injection of a mixture of LPL-Sm:CaF_2_ and PL-Sm:SrF_2_.	CaF_2_ or SrF_2_ nanofluorides doped with paramagnetic lanthanoides (Sm^3+^) and coated with phospholipids	Cohen et al., 2021 [107]
12	Generation of nanoparticles that encapsulate fluorinated ionic liquids in a nanoparticle shell. The shells were designed to be opened by acidic pH (H^+^), glutathione (GSH) and matrix metalloproteinases (MMPs) which resulted in the release of the ^19^F cargo (fluoroborate, difluoroacetate, and trifluoromethanesulfonate) which is associated with turning on the ^19^F signal. Due to the fact that all ^19^F tracer have a distinct chemical shift, it was possible to image GSH- and MMP-2- activity as well as acidic pH in subcutaneously implanted tumours in mice by in vivo multiplex ^19^F MRI.	Encapsulated fluorinated ionic liquids.	Zhu et al., 2022 [108]
13	Imaging of tumour-associated macrophages (TAM) in mouse models of gliomagenesis. I.v. injection of PFCs revealed that the ^19^F signal in the tumour was mainly due to ^19^F labelled TAMs. Application of PFCs with individual ^19^F spectra bevor and after radiotherapy revealed spatially and temporally distinct TAM niches (microglia and monocytes derived tumour macrophages).	*PFCs:* PFCE and PFTBH	Croci et al., 2022 [101]

**Abbreviations:** PFCs = perfluorocarbon nanoemulsions; PFOB = perfluorooctyl-bromide; PFCE = perfluoro-15-crown-5 ether; RARE = rapid acquisition with relaxation enhancement; PLGA = poly(lactic-co-glycolic) acid; PFO = perfluorooctane; TPFBME = 1,1,1-tris(perfluorotert-butoxymethyl)ethane; PFTBA = perfluorotributylamine; PERFECTA = suPERFluorinatEdContrasT Agent; PFCH = perfluoro-1,3,5-trimethylcyclohexane; PFTBH = perfluoro(tert-butylcyclohexane); FXIIIa = activated form of factor XIII.

### 4.2. Simultaneous Visualization of Monocytes/Macrophages and Thrombi

More recently, our group combined the active targeting of thrombi with the passive labelling of monocytes/macrophages to visualize thromboinflammation by multispectral ^19^F MRI in a mouse model that develops atherosclerotic plaques and plaque rupture after treatment with a high fat/cholesterol diet [103]. To detect FXIII activity and fibrin, we conjugated targeting peptides (α2AP and fbn) to the surface of PFCs that contain perfluoro-1,3,5-trimethylcyclohexane (PFCH) or perfluorooctyl-bromide (PFOB). Moreover, visualization of the phagocytic monocytes/macrophages was enabled with PFCs that contain perfluoro-15-crown-5 ether (PFCE). The perfluorocarbons PFCE, PFCH, and PFOB have individual spectral signals that are clearly separated and which can be selectively excited and detected (Figure 4A).

To verify the targeting specificity for thrombi, the surface of the PFCs was PEGylated (5 mol% DSPE-PEG_2000_). Incubation of RAW macrophages and murine monocytes from the blood with PFCE-PFCs, PEGylated PFCE-PFCs, ^α2AP^PFOB, and ^fbn^PFCH revealed that PFCE-PFCs displayed a strong cellular uptake whereas ^α2AP^PFOB, ^fbn^PFCH, and PEGylated PFCE-PFCs are taken up only weakly by phagocytic cells.

The specificity for thrombus imaging was first investigated by in vitro generated human thrombi that were incubated with ^α2AP^PFOB and ^fbn^PFCH. As control, the thrombi were also incubated with PFCs that were labelled with corresponding nonspecific control peptides. These experiments revealed that the binding of ^α2AP^PFOB and ^fbn^PFCH to in vitro generated thrombi was significantly stronger than the signals obtained from corresponding control-PFCs (Figure 4B). Further in vitro experiments did also show that ^α2AP^PFOB accumulate in early thrombi whereas ^fbn^PFCH binds to early as well as advanced thrombi (Figure 4C).

Finally, we utilized the combination of ^α2AP^PFOB, ^fbn^PFCH, and unmodified PFCE-PFCs to detect throminflammatory processes in HypoE mice, which spontaneously generate MI after treatment with a high-fat/cholesterol diet (7.5% cocoa butter, 15.8% fat, 1.25% cholesterol, and 0.5% sodium cholate). HypoE mice express a hypomorphic mutant form of ApoE (ApoE-Rh/h) and lack the scavenger receptor class B type I (SR-BI−/−). Previous studies have already shown that HypoE mice develop thromboinflammation and MI [109,110]. In this study, we observed that application of this PFC combination after three days of diet resulted in weak signals for monocytes/macrophages in the heart on day five. Of note, there were already strong ^19^F signals for all tracers found in the lungs at this time point (Figure 4D). However, on day 10 there were strong ^19^F signal for monocytes/macrophages, FXIIIa, and fibrin within the heart. Importantly, there was no major impact on heart function on day five, but after 10 days cardiac function was severely impaired. In particular, the function of the right ventricle was reduced which fits to the strong thromboinflammatory signals that were found in the lungs five days after onset of the diet. Interestingly, there was also an inverse correlation of the ^19^F signal on day five with cardiac function on day ten which indicates that this approach can also have prognostic value (Figure 4E).

Taken together, because the chemical shift range of ^19^F atoms is quite large (>300 ppm) and the ^19^F resonance frequencies vary depending on the chemical structure, it is possible to visualize different cells or structures via the selective excitation and detection of ^19^F tracer with individual spectral signals.

## 5. Conclusions and Outlook

In the recent years, ^19^F MRI has been established as a powerful tool to visualize phagocytic cells such as monocytes/macrophages or neutrophils under inflammatory conditions [111,112]. Moreover, functional modification of the surface of PFCs with polyethylene glycol and specific targeting ligands has strongly expanded the repertoire of the application for ^19^F MRI. Establishment of a platform for the active targeting of PFCs has enabled us to visualize FXIIIa activity [68] or activated platelets [70] independent of any phagocytic requirements. Moreover, attachment of ligands against CD177 on the surface of PFCs led to specific targeting of neutrophils and made it possible to follow their migration in vivo from the bone marrow into the infarcted heart [75]. More recently, we identified a peptide that targets cell surface CD63 which is strongly upregulated in activated cardiac fibroblasts as well as epicardial cells after myocardial infarction [73,74].

For imaging of thromboinflammation, it is necessary to visualize the inflammatory as well as the thrombotic components of this process. Multispectral ^19^F MRI is particularly well suited for this, because ^19^F has a broad spectral range of about ~400 ppm and there are several fluorine-containing compounds available with distinct individual signals that enable the detection of several target structures. However, the signal range that is currently exploited is in the range of only 30–40 ppm. Signals that differ by 5 ppm can be distinguished, so there is still a large range (up to ~70 theroretical ^19^F channels) for individual ^19^F signals. The design of molecules with magnetically identical ^19^F atoms and a distinct resonance frequency could strongly expand the repertoire of the structures and cells that can be visualized simultaneously by multispectral ^19^F MRI.

Most of the ^19^F MRI studies have been performed for preclinical research, but some studies have been conducted under clinical conditions. It was shown that it is possible to visualize the accumulation of monocytes/macrophages in the heart after myocardial infarction in pigs [49]. The same group also demonstrated the feasibility to image vessel wall inflammation in pigs after wire injury [113]. Of note, there has also been a small clinical study with ^19^F-labelled dendritic cells in adenocarcinoma patients [114]. There are still many challenges and pitfalls that must be overcome to enable the clinical application of ^19^F MRI which has been nicely summarized in a recent review article by van Heeswijk et al. [115]. These comprise the imaging hardware, the pulse sequences, the formulation and the dosing of the ^19^F tracer and also the specific clinical applications where ^19^F MRI can be beneficial for the medical personnel for diagnosis or decision making. ^19^F MRI is feasible at clinical field strength of 1.5 or 3T, but the MR systems require additional hardware for the acquisition of X nuclei and matching radiofrequency coils. PFCs require a regulatory approval which makes it necessary that they have to be manufactured under GMP (good manufacturing procedure) conditions. Furthermore, they have to be evaluated in phase 1 and 2 clinical trials to validate their safety and efficacy. One major issue with ^19^F MRI is the sensitivity because accumulation of ^19^F atoms in mM concentrations are normally necessary for detection. However, advanced techniques like compressed sensing have been shown to strongly reduce scan times by a factor of 4–8 which could decrease MRI measurement time during clinical applications or enhance the detection of low-intensity ^19^F signals [116,117]. Also, improved manufacturing processes (e.g., higher ^19^F content and larger particle size) or targeting specific receptors that leads to much better internalization could greatly increase the sensitivity. ^19^F signal detection could also be enhanced with the utilization of superconducting cryo-coils. For example, Waiczies et al. have compared ^19^F MRI of cryo-coils operating at room temperature and at 20K and found a 15-fold increased sensitivity for ^19^F in the brain of EAE (experimental autoimmune encephalomyelitis) mice [118].

Already, in early studies it has been observed that PFCs can cause side effects such as rare anaphylactoid responses that have been attributed to the utilization of the poloxamer pluronic F-68 as emulsifier [119,120], or transient flu-like symptoms [121]. The anaphylactic reactions were eliminated by emulsifying perfluorocarbons with purified lipids such as lecithin [119]. The low-level inflammatory reactions were related to the lipids and the particle size [122]. Of note, in more recent studies we did not observe any adverse impact of PFCs on cells or organs [43,68,75,123,124]. The active targeting of PFCs is currently based on PEGylation of PFCs to suppress of the cellular uptake by phagocytic cells (see Section 3). However, at least some variants of PEG are very immunogenic, which can result in adverse reactions that either neutralize the particles and accelerate their blood clearance or induce life threatening anaphylactic responses (CARPA = complement activation related pseudoallergy) [125]. Therefore, alternatives for these PEG-variants are required that overcome the adverse properties but still impair the phagocytic uptake by monocytes and macrophages.

Nevertheless, we believe that ^19^F MRI and, in particular, multispectral ^19^F MRI is a powerful molecular imaging tool that can be used to study multiple biological processes simultaneously with high specificity and quantitative accuracy. This technology is of high value for preclinical studies because it provides a large amount of additional locoregional information. Repetitive MRI measurements make it possible to follow and monitor the course of a disease (and its therapy) in the same animal over time, which enhances the reliability of the datasets and reduces animal numbers (3R principle). Furthermore, multispectral ^19^F MRI of thromboinflammation could also be important for clinical decision making. Imaging of inflammation and assessment of acute or chronic thrombi could help to decide if it is feasible to treat a specific patient with certain anti-inflammatory or thrombolytic drugs. Multispectral ^19^F MRI is non-ionizing, making it a safe alternative for in vivo monitoring of biological processes in human patients. Future advancements in multispectral ^19^F MRI, like an enhanced probe design for active targeting, computational image analysis can extend its application for molecular imaging and promote the monitoring of progression and therapy of various diseases. 

## Figures and Tables

**Figure 1 ijms-26-02462-f001:**
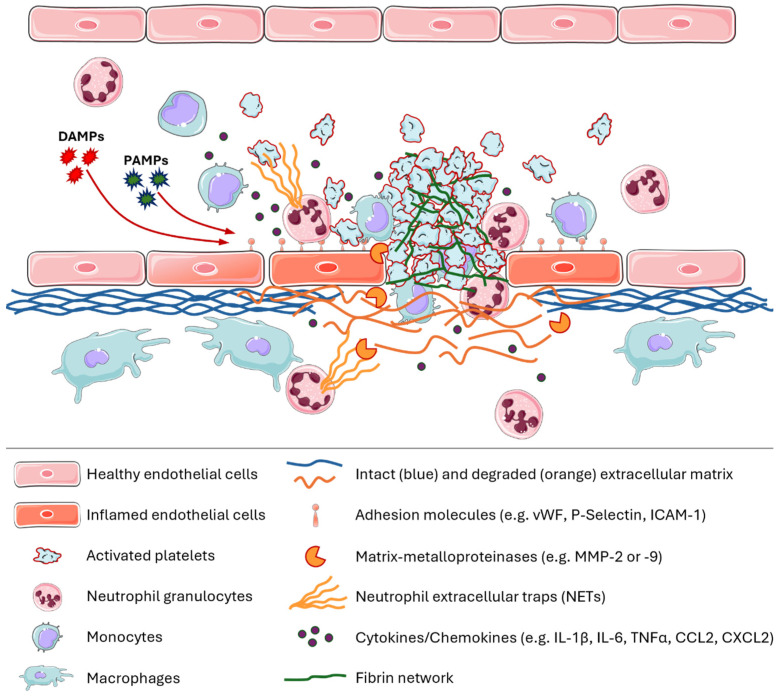
Schematic overview of the interaction between immune cells, platelets, and clotting molecules in thromboinflammation. Pathogen-associated molecular patterns (PAMPs) or damage-associated molecular patterns (DAMPs) can lead to endothelial dysfunction, resulting in the activation of endothelial cells that express adhesion molecules for platelets, monocytes, or neutrophils on the cell surface. Monocytes as well as neutrophils and platelets release various cytokines or chemokines to attract and activate other immune cells. The production of matrix metalloproteinases by platelets or neutrophils leads to the degradation of the extracellular matrix, which promotes thrombus formation. The thrombus, consisting of platelets but also monocytes and neutrophils, is further stabilized by a strong fibrin network. Neutrophils and monocytes also infiltrate into the inflamed area, where monocytes can differentiate into macrophages and lead to a generalized proinflammatory milieu. This figure was created with elements obtained from SMART-Servier Medical Art (https://smart.servier.com/ (accessed on 16 December 2024)).

**Figure 2 ijms-26-02462-f002:**
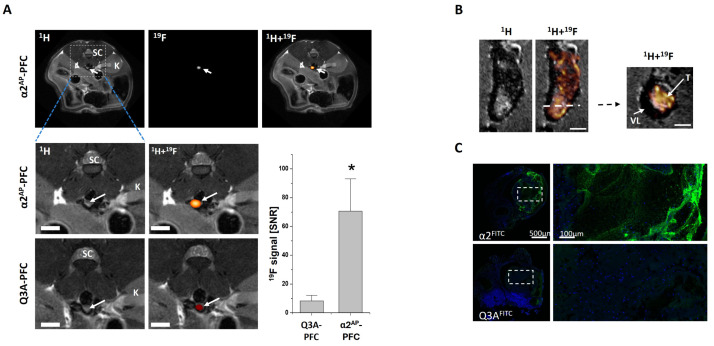
(**A**) Visualization of FeCl_3_-induced thrombi (arrow) in the inferior *vena cava* of male C57BL/6 mice by ^1^H/^19^F MRI. Upper panel: anatomical overview of the area of the thrombus location (arrow) (left). Corresponding ^19^F MRI (middle) and a merging of the ^1^H and ^19^F datasets which shows the site of the thrombus (right). The dashed lines in the lower part represent magnifications of ^1^H/^19^F MRI measurements of the thrombus area of mice which received α2^AP^-PFCs or Q3A-PFCs as control. The graph shows the signal to noise ratio of the ^19^F signal in the thrombus. Data are mean values ± SD of n = 8 (α2^AP^-PFCs), n = 6 (Q3A-PFCs), * = *p* < 0.05. K = kidney, and SC = spinal cord. (**B**) High-resolution ^1^H/^19^F MRI measurements (0.5 nL voxel size) of an excised venous thrombus that was fixed in paraformaldehyde and embedded in agarose. Please note that the ^19^F signal of the α2^AP^-PFCs is found as a patchy pattern within the thrombus. T = thrombus; VL = vessel lumen. (**C**) Fluorescence microscopy of excised thrombi after injection of PFCs functionalized with carboxyfluorescein-labelled α2^AP^ (upper panel) or Q3A (lower panel). The carboxyfluorescein is shown in green and nuclei of cells were counterstained with DAPI (DAPI = 4′,6-diamidino-2-phenylindole; blue). The dashed boxes represent the areas of the magnified images on the right. The scale bars represent 1 mm (**A**), 500 µm (**B**,**C**, left) or 100 µm (**C**, right). (The figure is a combination of elements derived from Figure 1 and Figure 3 of Temme et al. 2015 [68]; reprinted with permission).

**Figure 3 ijms-26-02462-f003:**
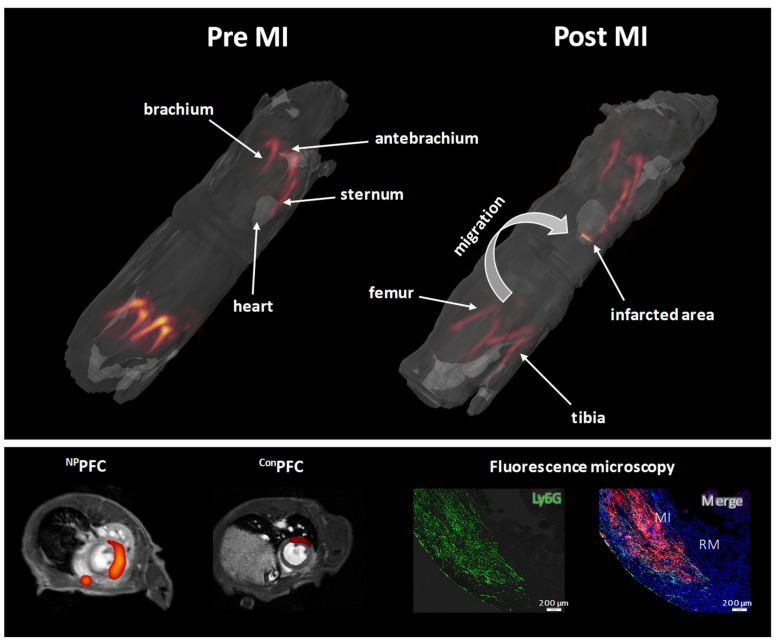
**Mapping the trafficking of neutrophils after myocardial infarction (MI) by ^1^H/^19^F MRI. Upper panel:** whole-body 3D ^1^H/^19^F MRI for systemic in vivo visualization of neutrophils pre- and post-MI. Anatomical ^1^H data were rendered transparent in grayscale with ^19^F data overlayed in orange/red; left: intravenous application of ^mNP^PFCs prior MI resulted in in situ labelling of neutrophils. Right: re-investigation 24 h post MI revealed a pronounced reduction in the ^19^F signals in the bone marrow and appearance of the ^19^F label in the infarcted heart. **Left lower panel:** midventricular axial ^1^H/^19^F MRI images post MI of mice that received ^NP^PFCs (left) or ^Con^PFCs that are functionalized with an unspecific control peptide (right). **Right lower panel:** fluorescence microscopic images of infarcted heart sections after injection of rhodamine-labelled ^NP^PFCs stained with anti-Ly6G antibodies to visualize neutrophils (left). Right: merging of the rhodamine- (red) with the Ly6G fluorescence (green) and cell nuclei stained with DAPI (4′,6-Diamidin-2-phenylindol; blue). (This figure was compiled with elements obtained from Bouvain et al., 2023 [75]; reprinted with permission).

**Figure 4 ijms-26-02462-f004:**
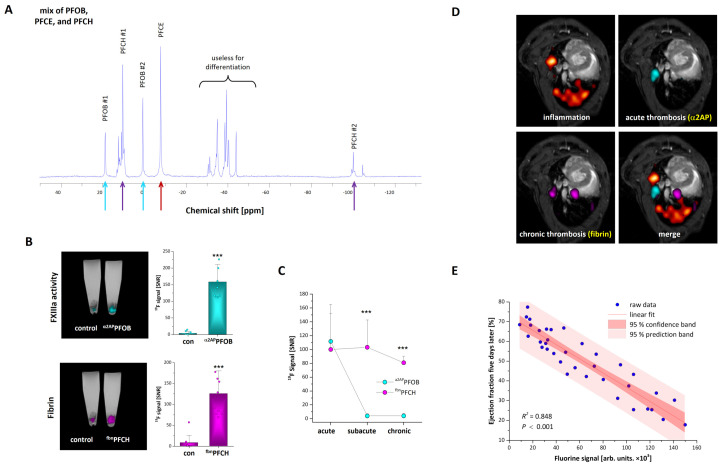
**Multispectral ^19^F MRI for visualization of inflammation and thrombosis:** (**A**) ^19^F spectrum of a combination of perfluoro-15-crown-5 ether (PFCE), perfluorooctyl-bromide (PFOB), and perfluoro-1,3,5-trimethylcyclohexane (PFCH). The arrows in the spectrum show individual ^19^F peaks than can be used for selective excitation and detection. (**B**) Longitudinal cross-sections of buffer-filled reaction tubes that contain ex vivo generated human thrombi labelled with ^α2AP^PFOB (upper), ^fbn^PFCH (lower), and corresponding control PFOB- and PFCH-PFCs. (**C**) Quantitative analysis of the ^19^F signal of ex vivo prepared acute, subacute, and chronic thrombi incubated with ^α2AP^PFOB and ^fbn^PFCH. (**D**) Multispectral ^19^F MRI images of the heart and the lungs of HypoE mice after five days of a high-fat/cholesterol diet. Conventional PFCE-PFCs (red = monocytes/macrophages), ^α2AP^PFOB [cyan = activated form of FXIII (FXIIIa)], and ^fbn^PFCH (magenta = fibrin). (**E**) Correlation between ^19^F signal at day x and the follow up of the cardiac function on day x + 5. *** = *p* < 0.001. Figure was modified from Flögel et al. [103] and reprinted with permission.

## Data Availability

No new data were created.

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
