# Peer review of "Imaging of Thromboinflammation by Multispectral ^19^F MRI"

_ijms, 2025, doi:10.3390/ijms26062462_

Round 1

Reviewer 1 Report

Comments and Suggestions for Authors

Sebastian Temme et al nicely summarized efforts around imaging of thromboinflammation by multispectral 19F MRI. The article is well written, but a few aspects need to be addressed. Despite 19F MRI being still a niche, the work could help to foster the dissemination of this very interesting technique.

Major Comments

-Lines 112-114: the authors state that “In general, these imaging techniques can be used to obtain information about structural changes in the tissue, to monitor the immune system or image thrombus formation [18–20].“. The references provided don’t seem adequate to me. More attention should be drawn to this, especially to PET/SPECT applications related to imaging inflammation related to thrombi at high sensitivity (e.g. Dogan ADA et al, BMC Res Notes 2024, doi: 10.1186/s13104-024-06820-w; Hammad B et al, J Nucl Cardiol 2017, doi: 10.1007/s12350-016-0766-y; Kessinger CW et al, Circ Cardiovasc Imaging 2021, doi: 10.1161/CIRCIMAGING.120.011898; Lin Q et al, Clin Nucl Med 2023, doi: 10.1097/RLU.0000000000004803; Pervaiz MH et al, J Nucl Cardiol 2018, doi: 10.1007/s12350-017-0826-y; Raynor WY et al Diagnostics (Basel) 2021, doi: 10.3390/diagnostics11122234; Shimizu Y, Kuge Y, Nucl Med Mol Imaging 2016, doi: 10.1007/s13139-016-0418-9; Stacy MR, Curr Cardiovasc Imaging Rep 2019, doi: 10.1007/s12410-019-9491-7). This part should be rewritten in more detail, putting advantages/limitations of each technique in evidence, with appropriate references.

-Lines 156-158: instead of limiting the application of SPIOs to one reference, it would be appropriate to additionally call the attention of the reader to relevant reviews dealing with the subject (e.g. Beckmann N et al, doi: 10.1002/wnan.16; Dadfar SM et al, doi: 10.1016/j.addr.2019.01.005; Tsampasian V et al, doi: 10.1186/s12880-021-00695-0; Ugga L et al, doi: 10.1016/j.jneumeth.2018.06.008; Stoll G, Bendszus M, doi: 10.1097/WCO.0b013e328337f4b5; Wáng YX, Idée JM, doi: 10.21037/qims.2017.02.09).

-Lines 200-202: the authors should also mention that very early inflammatory events (4 h, 6 h) in the lungs following LPS administration could be detected by 1H and/or 3He MRI (e.g. Olsson LE et al, doi: 10.1002/jmri.21728; Quintana HK et al, doi: 10.1152/ajplung.00303.2005).

-Lines 216-227: appropriate references for the different functionalization methods should be provided here.

-Lines 228-234: appropriate references for PEGylation method with the aim to facilitate the internalization of the nanoparticles need to be provided.

-Lines 277-282: since this is the first example of a targeted PFC in the article, representative images of thrombi obtained in vivo with corresponding histological confirmation should be presented.

-Line 333: please state the percentage of neutrophils expressing CD177 in humans.

-Lines 341-342: please state the approximate uptake by monocytes or lymphocytes.

-Line 405: the acronym to which PERFECTA stands for needs to be specified.

-Lines 457-459: the inverse correlation between the 19F signal on day five and the cardiac function on day 10 should be added to figure 3.

-Section 5: Limitations of 19F MRI should be briefly commented. Moreover, potential side effects of such nanoconjugates need to be mentioned. Although already addressed in reference 91, two or three sentences about challenges to bring this technology to the clinics would be welcome to facilitate reading the article.

Minor Comments

-Line 146: would rather write “migration of immune cells”.

-Line 242: suggest rewriting the sentence to “Active targeting of PFCs has enabled …”.

-Lines 254-255: suggest rewriting the sentence to “… in thrombi in T1‐weighted proton MRI images.”.

-Line 373: should be “… allows the simultaneous …”.

-Lines 387-388: suggest rewriting the sentence to “… then reintroduce the cells into an organism to track them in vivo [71].”.

-Line 393: since the abbreviation is mentioned for the first time in the text, suggest rewriting sentence to … “PFC‐loaded nanoprobes (perfluorooctyl‐bromide (PFOB) and perfluoro‐15‐crown‐5 ether (PFCE) labeled PFCs) for in vivo …”.

Author Response

#Reviewer 1:

Comments and Suggestions for Authors

Sebastian Temme et al nicely summarized efforts around imaging of thromboinflammation by multispectral 19F MRI. The article is well written, but a few aspects need to be addressed. Despite 19F MRI being still a niche, the work could help to foster the dissemination of this very interesting technique.

Thank you very much for the positive evaluation of our manuscript and the helpful comments and suggestions.

Major Comments

-Lines 112-114: the authors state that “In general, these imaging techniques can be used to obtain information about structural changes in the tissue, to monitor the immune system or image thrombus formation [18–20].“. The references provided don’t seem adequate to me. More attention should be drawn to this, especially to PET/SPECT applications related to imaging inflammation related to thrombi at high sensitivity (e.g. Dogan ADA et al, BMC Res Notes 2024, doi: 10.1186/s13104-024-06820-w; Hammad B et al, J Nucl Cardiol 2017, doi: 10.1007/s12350-016-0766-y; Kessinger CW et al, Circ Cardiovasc Imaging 2021, doi: 10.1161/CIRCIMAGING.120.011898; Lin Q et al, Clin Nucl Med 2023, doi: 10.1097/RLU.0000000000004803; Pervaiz MH et al, J Nucl Cardiol 2018, doi: 10.1007/s12350-017-0826-y; Raynor WY et al Diagnostics (Basel) 2021, doi: 10.3390/diagnostics11122234; Shimizu Y, Kuge Y, Nucl Med Mol Imaging 2016, doi: 10.1007/s13139-016-0418-9; Stacy MR, Curr Cardiovasc Imaging Rep 2019, doi: 10.1007/s12410-019-9491-7). This part should be rewritten in more detail, putting advantages/limitations of each technique in evidence, with appropriate references.

Thank you very much for addressing this point. We agree with the reviewer that the references do not perfectly fit and that the description of other techniques is rather short. We therefore expanded this part and included a paragraph to describe/summarize some of the PET/SPECT applications for imaging of inflammation and thrombosis (Page 3, Lines 114-129).

-Lines 156-158: instead of limiting the application of SPIOs to one reference, it would be appropriate to additionally call the attention of the reader to relevant reviews dealing with the subject (e.g. Beckmann N et al, doi: 10.1002/wnan.16; Dadfar SM et al, doi: 10.1016/j.addr.2019.01.005; Tsampasian V et al, doi: 10.1186/s12880-021-00695-0; Ugga L et al, doi: 10.1016/j.jneumeth.2018.06.008; Stoll G, Bendszus M, doi: 10.1097/WCO.0b013e328337f4b5; Wáng YX, Idée JM, doi: 10.21037/qims.2017.02.09).

We agree with the reviewer that we should include some more references that describe the properties and applications of SPIOs. We revised and extended this part and included the literature that was suggested by the reviewer (Page 5, Lines 167-173).

-Lines 200-202: the authors should also mention that very early inflammatory events (4 h, 6 h) in the lungs following LPS administration could be detected by 1H and/or 3He MRI (e.g. Olsson LE et al, doi: 10.1002/jmri.21728; Quintana HK et al, doi: 10.1152/ajplung.00303.2005).

Thanks for mentioning this interesting point. We added a short paragraph to describe that early pulmonary inflammation can also be imaged by 1H and 3He MRI (Page 6, Lines 219-222).

-Lines 216-227: appropriate references for the different functionalization methods should be provided here.

As suggested by the reviewer, we added references for the functionalization methods. Moreover, we revised the whole section to enhance the readability (see below & Page 6, Lines 238-244). 

References:

Hermanson, G.T. Chapter 3 - The Reactions of Bioconjugation. In Bioconjugate Techniques (Third Edition); Hermanson, G.T., Ed.; Academic Press: Boston, 2013; pp. 229–258 ISBN 978-0-12-382239-0.

Lesch, H.P.; Kaikkonen, M.U.; Pikkarainen, J.T.; Ylä-Herttuala, S. Avidin-Biotin Technology in Targeted Therapy. Expert Opinion on Drug Delivery 2010, 7, 551–564, doi:10.1517/17425241003677749.

Al-Seragi, M.; Chen, Y.; Duong van Hoa, F. Advances in Nanobody Multimerization and Multispecificity: From in Vivo Assembly to in Vitro Production. Biochem Soc Trans 2025, 53, BST20241419, doi:10.1042/BST20241419.

-Lines 228-234: appropriate references for PEGylation method with the aim to facilitate the internalization of the nanoparticles need to be provided.

Please apologize that we missed to provide the necessary references. We revised this part and included appropriate literature (See below & Page 6, Lines 248-251).

References:

Partikel, K.; Korte, R.; Stein, N.C.; Mulac, D.; Herrmann, F.C.; Humpf, H.-U.; Langer, K. Effect of Nanoparticle Size and PEGylation on the Protein Corona of PLGA Nanoparticles. Eur J Pharm Biopharm 2019, 141, 70–80, doi:10.1016/j.ejpb.2019.05.006.

Owens, D.E.; Peppas, N.A. Opsonization, Biodistribution, and Pharmacokinetics of Polymeric Nanoparticles. International Journal of Pharmaceutics 2006, 307, 93–102, doi:10.1016/j.ijpharm.2005.10.010.

-Lines 277-282: since this is the first example of a targeted PFC in the article, representative images of thrombi obtained in vivo with corresponding histological confirmation should be presented.

Thank you very much for the hint. We agreed and we added an additional figure that shows the visualization of early thrombi by 1H/19F MRI. The figure displays the in vivo imaging of FeCl3-induced deep venous thrombi after application of specific α2AP-PFCs as well as Q3A-PFCs as control. We also included the high resolution 1H/19F MRI and the fluorescence microscopy of excised thrombi which confirmed that the α2AP-PFCs are indeed located within the thrombus. We added this as new Figure 2 (Page 8, Lines 309-324).

-Line 333: please state the percentage of neutrophils expressing CD177 in humans.

We added the information that in most human individuals, approximately 50-60 % of the neutrophils are positive for CD177, although the range is from 0-100 %. (Page 9, lines 373-380).

-Lines 341-342: please state the approximate uptake by monocytes or lymphocytes.

Thank you very much for making us aware that additional information would be helpful. There are some differences in the uptake of NPPFCs by lymphocytes and monocytes which are dependent on the cell type, but also on the specific situation (e.g. blood vs. inflammatory lesion) or the technology. In general, there is an increased uptake by neutrophils in the range of ~5x – 50x. We now included this information in the revised manuscript (Page 9, Lines 385-390).

-Line 405: the acronym to which PERFECTA stands for needs to be specified.

Sorry that we missed to explain where the word PERFECTA comes from. It is not an acronym, but it is derived from the words „suPERFluorinatEd ContrasT Agent“. We added this explanation to the text (Page 11, lines 454-455).

-Lines 457-459: the inverse correlation between the 19F signal on day five and the cardiac function on day 10 should be added to figure 3.

We agree with the reviewer that the correlation between the 19F signal on day 5 and the functional outcome on day 10 is an interesting information for the reader. We included this graph to figure 4 (Page 15, lines 522-523).

-Section 5: Limitations of 19F MRI should be briefly commented. Moreover, potential side effects of such nanoconjugates need to be mentioned. Although already addressed in reference 91, two or three sentences about challenges to bring this technology to the clinics would be welcome to facilitate reading the article.

Thank you very much for addressing these points. We fully revised this part and added paragraphs about (i) limitations of 19F MRI and challenges for clinical translation and (ii) side effects of nanoconjugates (Page 16, Lines 565-599).

Minor Comments

-Line 146: would rather write “migration of immune cells”.

Thanks, corrected (Page 4, Lines 161-162)

-Line 242: suggest rewriting the sentence to “Active targeting of PFCs has enabled …”.

Corrected (Page 6, Lines 260-261).

-Lines 254-255: suggest rewriting the sentence to “… in thrombi in T1‐weighted proton MRI images.”.

We agree, your suggestion makes it more specific. (Page 7, Lines 280-281)

-Line 373: should be “… allows the simultaneous …”.

The sentence was changed in accordance to the suggestion of the reviewer (Page 10, Lines 421-422).

-Lines 387-388: suggest rewriting the sentence to “… then reintroduce the cells into an organism to track them in vivo [71].”.

We agree, corrected (Page 11, Lines 435-436).

-Line 393: since the abbreviation is mentioned for the first time in the text, suggest rewriting sentence to … “PFC‐loaded nanoprobes (perfluorooctyl‐bromide (PFOB) and perfluoro‐15‐crown‐5 ether (PFCE) labeled PFCs) for in vivo …”.

Thank you for making us aware that we missed to explain the abbreviation. We revised this sentence (Page 11, Lines 441-442).

Reviewer 2 Report

Comments and Suggestions for Authors

The article entitled “Imaging of thromboinflammation by multispectral 19F MRI” is a review about the imaging method such as the MRI based on fluorine 19 that is able to diagnose the thromboinflammation through the targeting of the immune and hemostatic cells such as neutrophil granulocytes, monocytes, macrophages, platelets.  The topic is clinically important as the thromboinflammation is a pathogenetic factor of cardiovascular disease. The authors exhaustively describe either the pathophisiology of the thromboinflammation, the technical caharteristics of 19F MRI” , and the the published articles on the argument. Although this article is a review, the authors have experience about the 19F MRI in animal and human experimental studies and have published their findings in international journals. Therefore, I think that this article is suitable for publication in its current version.

Author Response

#Reviewer 2:

Comments and Suggestions for Authors

The article entitled “Imaging of thromboinflammation by multispectral 19F MRI” is a review about the imaging method such as the MRI based on fluorine 19 that is able to diagnose the thromboinflammation through the targeting of the immune and hemostatic cells such as neutrophil granulocytes, monocytes, macrophages, platelets.  The topic is clinically important as the thromboinflammation is a pathogenetic factor of cardiovascular disease. The authors exhaustively describe either the pathophisiology of the thromboinflammation, the technical caharteristics of 19F MRI” , and the the published articles on the argument. Although this article is a review, the authors have experience about the 19F MRI in animal and human experimental studies and have published their findings in international journals. Therefore, I think that this article is suitable for publication in its current version.

Thank you very much for this positive evaluation of our manuscript.

Reviewer 3 Report

Comments and Suggestions for Authors

Temme et al summarized the background of 19F MRI for inflammation imaging, active targeting approaches to visualize thrombi and specific immune cells, introduce studies about multispectral 19F MRI. This manuscript is well prepared and I have only minor comments.

1.     Recommend describe the effect size of treatment effect, e.g. with IL-1β blockade at lines 98-100.

2.     Line 334, Typo “C57BL/9J”.

3.     Recommend add a sentence in the end of each section to summarize the unique biological findings/application potentials/inferences of (multispectral) 19F MRI in each section.

4.     Add explanations for abbreviations under Table 1.

Author Response

#Reviewer 3:

Comments and Suggestions for Authors

Temme et al summarized the background of 19F MRI for inflammation imaging, active targeting approaches to visualize thrombi and specific immune cells, introduce studies about multispectral 19F MRI. This manuscript is well prepared and I have only minor comments.

1.Recommend describe the effect size of treatment effect, e.g. with IL-1β blockade at lines 98-100.

We agree with the reviewer that some details on the clinical outcome would be interesting. Unfortunately, the authors of this study did not provide any information about the effect size. We therefore added information about the primary cardiovascular clinical outcomes of the group that received 150 mg of canakinumab which showed the most beneficial effects. (Page 3, Lines 96-98)

2.Line 334, Typo “C57BL/9J”.

Thank you very much. We corrected this to C57BL/6J (Page 9, Line 379).

3. Recommend add a sentence in the end of each section to summarize the unique biological findings/application potentials/inferences of (multispectral) 19F MRI in each section.

Thank you very much for this point. We evaluated the manuscript very carefully with regard to this specific suggestion. We do not think that a summarizing sentence is necessary for all sections. However, we added a brief summary to the following paragraphs: (i) 3. Active targeting of perfluorocarbon nanoemulsions (Page 7, Lines 264-271) & (ii) 4.3 Simultaneous visualization of monocytes/macrophages and thrombi (Pages 14, Lines 518-521)

4. Add explanations for abbreviations under Table 1.

Sorry that we missed to explain the abbreviations in the table. In the revised manuscript, we listed all abbreviations and their full names under table 1. (Page 13-14, Lines 469-475).

Round 2

Reviewer 1 Report

Comments and Suggestions for Authors

Points raised were well addressed, thanks. Nice review.